# Prevalence and Risk Factors Associated with Multidrug Resistance and Extended-Spectrum *β*-lactamase Producing *E. coli* Isolated from Healthy and Diseased Cats

**DOI:** 10.3390/antibiotics12020229

**Published:** 2023-01-20

**Authors:** Mahmoud Fayez, Ahmed Elmoslemany, Ahmad A. Al Romaihi, Abdulfattah Y. Azzawi, Abdullah Almubarak, Ibrahim Elsohaby

**Affiliations:** 1Al-Ahsa Veterinary Diagnostic Lab, Ministry of Environment, Water and Agriculture, Al-Ahsa 31982, Saudi Arabia; 2Department of Bacteriology, Veterinary Serum and Vaccine Research Institute, Ministry of Agriculture, Cairo 131, Egypt; 3Hygiene and Preventive Medicine Department, Faculty of Veterinary Medicine, Kafrelsheikh University, Kafr El-Sheikh 33516, Egypt; 4National Center for the Prevention and Control of Plants and Animal Diseases, Riyadh 11195, Saudi Arabia; 5Ibrize Life Sciences, Dammam 32242, Saudi Arabia; 6Department of Animal Medicine, Faculty of Veterinary Medicine, Zagazig University, Zagazig City 44511, Egypt; 7Department of Infectious Diseases and Public Health, Jockey Club College of Veterinary Medicine and Life Sciences, City University of Hong Kong, Hong Kong SAR 999077, China; 8Centre for Applied One Health Research and Policy Advice (OHRP), City University of Hong Kong, Hong Kong SAR 999077, China

**Keywords:** cats, *E. coli*, ESBL, risk factors, antimicrobial resistance, multidrug resistance

## Abstract

Household cats have been identified as potential antimicrobial resistance (AMR) reservoirs, and the extended-spectrum *β*-lactamases (ESBL) producing *E. coli* circulating among cats has been more frequently reported globally, but the factors linked to its colonization remain poorly understood. Thus, the objectives of this study were to determine *E. coli* shedding and the occurrence of multidrug resistant (MDR)- and ESBL-producing *E. coli*, as well as to determine risk factors associated with colonization of MDR and ESBL-producing *E. coli* isolated from both healthy and diseased cats in the Eastern Province of Saudi Arabia. In a cross-sectional study, 2000 swabs were collected from five anatomical regions (anus, skin, ear canal, nares, and conjunctival sac) of 209 healthy and 191 diseased cats that were admitted to a veterinary clinic in the Eastern Province of Saudi Arabia. In addition, each cat owner filled out a questionnaire about their cat’s demographics, management, health status, and antimicrobial usage. *E. coli* was detected in 165 (41.3%) of all cats, including 59 (28.2%) healthy and 106 (55.5%) diseased cats. In total, 170 *E. coli* isolates were found in healthy (35.3%) and diseased (64.7%) cats. Susceptibility testing revealed that 123 (72.4%) of the *E. coli* isolates were resistant to at least one of the tested antimicrobials. Overall, 17.6% (30/170) of *E. coli* isolates were MDR, with 10 (5.9%) and 20 (11.8%) isolates found in healthy and diseased cats, respectively. However, only 12 (7.1%) *E. coli* isolates were resistant to cefotaxime and harbored the *bla*_CTX-M_ gene (ESBL-producer), with seven (4.1%) in healthy and five (2.9%) in diseased cats. Risk factor analysis showed that the odds of MDR and ESBL-producing *E. coli* were (20 and 17) and (six and eight) times higher when the family and cats were previously treated with antimicrobials, respectively. The presence of a child in the cat’s family was also linked to an increased risk of MDR *E. coli* colonization (OR = 3.4). In conclusion, a high frequency of MDR and ESBL-producing *E. coli* was detected among healthy and diseased cats in Saudi Arabia, raising concerns about transmission to humans and supporting the need of a “One Health” approach to address the potential threats of cats as AMR reservoirs.

## 1. Introduction

Antimicrobials are important therapeutic agents for infectious bacterial diseases in companion animals. Antimicrobial substance efficacy loss can have serious consequences for animal health and welfare [1]. Antimicrobial resistance (AMR) in companion animals is a complicated topic that is becoming increasingly important due to patient factors and public health concerns [2]. A unique feature of AMR in companion animals is that their close contact with humans creates favorable conditions for interspecies transmission of multidrug resistant bacteria (MDR) [3]. An additional risk factor for the emergence and spread of AMR is the use of antimicrobials crucial to human health in companion animals [4].

*Escherichia coli* (*E. coli*) is a Gram-negative intestinal commensal that is frequently isolated from a broad range of infection sites in both animals and humans [5]. *E. coli* variants were classified based on their virulence properties into several pathotypes including Shiga toxin-producing *E. coli* (STEC), Enteroinvasive *E. coli* (EIEC), Enteropathogenic *E. coli* (EPEC), Enterotoxigenic *E. coli* (ETEC), Enterohaemorrhagic *E. coli* (EHEC), and Enteroaggregative *E. coli* [5]. The majority of the virulence genes, including the *stx1*, *stx2*, *eaeA*, and *hlyA*, are important for *E. coli* pathogenicity and are linked to diarrhea in both animals and humans [5]. Selection of appropriate antimicrobials is critical for effective treatment of these infections and subsequently to reduce the risk of the emergence of AMR in commensal and pathogenic *E. coli* [6]. However, antimicrobial misuse led to the development of MDR and extended-spectrum *β*-lactamase (ESBL) producing *E. coli* in healthy and diseased companion animals [7,8]. Dissemination of ESBL-producing *E. coli* among companion animals resulted in reduction in efficacy or failure of the antimicrobial therapy [9]. Furthermore, the presence of ESBL genes in commensal *E. coli* in healthy animals could transferred horizontally to pathogenic strains [10] and could also spread resistance to humans through fecal–oral contamination [11].

In Saudi Arabia, cats are among the most common household pets [12]. The close contact between cats and their owners provides favorable conditions for transmission of zoonotic pathogens, either directly (e.g., through petting, licking, or physical injuries) or indirectly (e.g., through contamination of food or the environment) [13]. ESBL genes have been found in commensal and clinical *E. coli* isolates recovered from cats and/or their owners worldwide [14], suggesting that ESBL-producing *E. coli* can spread in both directions from humans and animals [15,16].

Despite the fact that numerous studies have found ESBL-producing *E. coli* in companion animals all over the world [17,18,19], to the authors’ knowledge, no data on AMR profiles of *E. coli* isolates from cats in Saudi Arabia have been published yet. Furthermore, studies on the risk factors associated with ESBL-producing *E. coli* colonization in cats are limited. Therefore, the objectives of this study were to (1) investigate *E. coli* shedding from various anatomical locations in healthy and diseased cats; (2) determine the frequency of MDR and ESBL-producing *E. coli* isolated from healthy and diseased cats; (3) investigate the presence of some selected virulence genes to identify the pathotypes of ESBL-producing *E. coli* isolates; and (4) determine the potential risk factors linked to colonization of MDR and ESBL-producing *E. coli* in both healthy and diseased cats in the Eastern Province of Saudi Arabia.

## 2. Results

### 2.1. Cat Population

A total of 400 cats were sampled, with 52.3% (209/400) apparently healthy and admitted for vaccination and/or grooming. However, the remining 191 (47.7%) cats were admitted for clinical examination due to digestive distress (81/400), ear discharge (61/400), eye discharge (59/400), respiratory distress (39/400), and skin wound and/or abscess (64/400). The sampled cats were 56.7% females and 43.3% males. The study population included five different cat breeds: Persian (40.5%), Siamese (28%), Himalayan (14.8%), Birman (13.2%), Egyptian Mau (2.5%), and Arabian Mau (1%). The characteristics of the studied cat population are detailed in Appendix A.

### 2.2. Prevalence of E. coli Isolates 

Among all cats (*n* = 400), 165 (41.3%) were found to carry *E. coli* including 59 (28.2%) healthy and 106 (55.5%) diseased. A total of 170 *E. coli* isolates (including isolates recovered from two different anatomical locations in five cats) were found in healthy (35.3%) and diseased (64.7%) cats. There were no significant differences in the distribution of *E. coli* isolates between healthy and diseased cats, cat breeds, or sex (*p* > 0.33). The prevalence of *E. coli* isolates in healthy and diseased cats based on anatomical locations is presented in Table 1.

### 2.3. Antimicrobial Susceptibility Testing 

Susceptibility testing revealed that 123 (72.4%) of the *E. coli* isolates showed resistance to at least one of the tested antimicrobials (Figure 1). The antimicrobial susceptibility test showed that 68.3% and 45.5% of resistant *E. coli* isolates were found in healthy and diseased cats, respectively. *E. coli* isolates resistant to AMP (53.5%) was the highest, followed by resistance to SXT (22.9%); however, none of the isolates were MEM resistant (Figure 2A). *E. coli* isolates found in healthy cats exhibited resistance patterns comparable to isolates recovered from diseased cats. Furthermore, the frequency of *E. coli* isolates resistant to AMP (odds ratio (OR): 2.6) was significantly higher in healthy cats than in diseased cats; however, no differences (*p* > 0.05) were found between diseased and healthy cats for the other antimicrobials. The frequency of resistant *E. coli* isolates recovered from various anatomical locations in healthy and diseased cats is depicted in Figure 2B. The MAR index of resistant *E. coli* isolates ranged from 0.09 to 0.64 with 24.4% of the isolates having an AMR index >0.2. AMR index of *E. coli* isolates from healthy (average = 0.21; range: 0.09–0.64) and diseased (average = 0.22; range: 0.09–0.55) cats was not significantly different (Figure 3).

### 2.4. MDR and ESBL-Producing E. coli 

Overall, 17.6% (30/170) of *E. coli* isolates tested positive for MDR, with 10 (5.9%) and 20 (11.8%) isolates found in healthy and diseased cats, respectively (Table 2). MDR isolates were resistant to three to six different classes of antimicrobials. Furthermore, among the 170 *E. coli* isolates, only 12 (7.1%) were resistant to CTX and harbored the *bla*_CTX-M_ gene (ESBL-producer) with seven (4.1%) in healthy and five (2.9%) in diseased cats (Table 2). Four different ESBL genes were found (*bla*_CTX-M-1 and 15_, *bla*_TEM_ and *bla*_SHV_), and *bla*_CTX-M-15_ predominated (66.7%, 8/12), followed by *bla*_TEM_ (50%, 6/12) (Table 3). Furthermore, one or more virulence genes were found in 41.7% (5/12) of the ESBL-producing *E. coli* isolates. The *eaeA* virulence gene was the most detected (5/12), followed by *stx2* and *hlyA* (2/12). None of the ESBL-producing *E. coli* isolates possessed the virulence gene *stx1* (Table 3).

### 2.5. Risk Factors for MDR and ESBL-Producing E. coli 

Univariable analysis (*p* < 0.25) revealed a positive association between both MDR and ESBL-producing *E. coli* and a number of family characteristics such as antimicrobial use, previous antimicrobial use for the cat, having a child at home, and cat food type. However, only MDR *E. coli* was positively linked to the presence of acne, current use of antimicrobials for cats, and reason for visiting the veterinary clinic (Table 4). On the other hand, when females care for the cat, that reduces the risk of MDR and ESBL compared to males. 

The results of multivariable analysis for factors associated with having MDR and ESBL-producing *E. coli* are shown in Table 5. The odds of MDR and ESBL were (20 and 17) and (six and eight) times higher when the family and cats were previously treated with antimicrobials, respectively. Furthermore, feeding a cat raw food or home-available food increases the odds of having MDR (nine and six times) and ESBL-producing *E. coli* (60 and 12 times) compared to feeding a cat dry food, respectively. The presence of a child in the cat’s family was also associated with an increased risk of MDR *E. coli* (OR = 3.4). Finally, cats in the care of females had lower odds of having MDR *E. coli* (OR = 0.2) than cats in the care of males.

## 3. Discussion

The number of animals kept as pets has increased significantly in recent decades; approximately 223 million pets are owned globally today [20]. Cats are one of the most common household pets, providing their owners with joy and companionship. However, the increased interaction between household cats and their owners creates favorable conditions for resistant pathogens transmission through direct and indirect contact [13,21]. Several studies have investigated the public health risks associated with the transfer of antimicrobial-resistant bacteria from cats [1,22]; however, studies investigated the risk factors associated with antimicrobial-resistant bacteria colonization are scarce. In the present study, 41.3% of cats were *E. coli* carriers, which is consistent with the *E. coli* isolation rates reported in Canada [23] and Hong Kong [24]. However, it was higher than the isolation rate (8.7%) reported in cats in South Korea [25]. This variation could be attributed to the number of cats in each study, sample type, sampling site, and cat health status. In this study, *E. coli* was recovered from 28.2% and 55.5% of apparently healthy and diseased cats, respectively. A similar isolation rate (52%) was observed in diseased cats in Italy [26], whereas a higher isolate rate (45.6%) was found among apparently healthy cats in China [19].

Antimicrobial resistance is an emerging and growing threat among many clinically relevant bacteria including *E. coli*. The high frequency of resistant *E. coli* (72.4%) found in the present study against antimicrobials commonly used in small animals and humans is an alarming finding. The results of this study are consistent with the high frequency of resistance to AMP and SXT reported for feline *E. coli* isolates in other studies [27,28]; however, the resistant rate to AMC (6.4%) was lower than that (15–100%) found in these studies. Moreover, all *E. coli* isolates were susceptible to MEM, suggesting the absence of carbapenems resistance among our isolates which is consistent with previous studies in cats from Australia [27] and China [19].

The MDR bacteria isolated from food-producing and companion animals have become an emerging problem [29,30]. In the present study, 17.6% of *E. coli* isolates were identified as MDR. Most studies reported a higher percentage of MDR *E. coli* in companion animals such as 66.8% in Poland [31], 56% in USA [18], 43.3% in Japan [32], and 23.8% in South Korea [28]. However, other studies have reported a lower percentage of MDR *E. coli* in cats such as 11.7% in Australia [27]. The variations in the percentages of MDR *E. coli* between studies could be explained by differences in the criteria used to classify isolates as MDR. The criteria developed by Magiorakos, et al. [33] were used to classify our isolates. Furthermore, the origin of the MDR isolates could explain the variation. Isolates recovered from healthy animals may not have been exposed to antimicrobials and thus demonstrate a low level of resistance. In this study, MDR isolates were found in both healthy and diseased cats; however, the percentage of MDR *E. coli* recovered from diseased (11.8%) was double that isolated from healthy cats (5.9%).

Many studies in the last decade have reported the spread of ESBL-producing *E. coli* from clinical isolates in cats [34,35]. Furthermore, it is now known that healthy animals can harbor antimicrobial resistant pathogens, including ESBL-producing *E. coli* [29,36]. In our study, the overall prevalence of ESBL-producing *E. coli* was 7.1%. A comparable prevalence has also been reported in UK (7%) [37] and Switzerland (7.5%) [38]. This prevalence, however, was higher than the 3.7% and 2% reported in companion animals in France [39] and Netherland [40]. Other studies have reported 0% of cats tested positive for ESBL-producing *E. coli* [29,41]. One of the possible explanations for this difference is the antimicrobial use strategy used by veterinarians in different countries. In this study, healthy cats (4.1%) had a higher rate of ESBL-producing *E. coli* than diseased cats (2.9%). Similarly, several studies have found ESBL-producing *E. coli* in healthy cats and dogs, but the prevalence is higher in diseased animals [42,43]. Although ESBL-producing *E. coli* was isolated from various anatomical regions, anus swabs were the most common source of ESBL-producing *E. coli* in our study, which is consistent with many other studies [42,44]. This is not surprising given that *E. coli* is frequently isolated from the digestive tract.

A diversity of *bla*_ESBL_ genes were reported in this study and are similar to those found in both humans [45] and food-producing animals [46] in Saudi Arabia. The predominant resistance gene found in the present study is *bla*_CTX-M-15_, which is consistent with previous studies in companion animals [37,44,47]. However, the *bla*_CTX-M-1_ was reported as the predominant in other studies [43,48], and the *bla*_CTX-M-55_ was detected in pets in mainland China [49]. 

In the present study, virulence associated genes (*eaeA*, *stx2*, and *hlyA*) were found in ESBL-producing *E. coli* isolated from healthy and diseased cats. *E. coli* is an opportunistic pathogen that lives in the intestinal microbiota of animals and humans. In this study, the molecular detection of the *eaeA*, *stx2*, and *hlyA* virulence genes suggested the dissemination of enteropathogenic and enterohemorrhagic *E. coli* in the investigated cats. Similar studies have reported the isolation of pathogenic *E. coli* from cats [50,51].

The identification of factors associated with MDR and ESBL-producing *E. coli* in pets would be valuable for controlling transmission between humans and animals. Previous studies showed that pet contact was related to ESBL carriage in humans [52,53]. Therefore, cats carrying ESBL-producing *E. coli* represent a potential risk to human health since they live closely together. This study found a higher risk of MDR and ESBL-producing *E. coli* in cats and family members who had previously received antibiotic treatment. This finding is in line with previous findings in humans [54] and companion animals [27,55]. A study on cats also revealed that prior antimicrobial treatment has a significant influence on the likelihood of *E. coli* isolate exhibiting MDR [56]. There was a positive association between MDR and having children at home. Children handle animals and touch their faces or mouths more frequently than adults, which increase the risk of disease transmission. The current study also showed feeding cats raw food was associated with higher risk of MDR and ESBL-producing *E. coli* infection compared to cats fed on dry food. A cohort study on 36 household cats showed strong association between feeding raw pet food and ESBL shedding [57]. The same study also isolated ESBL *Enterobacteriaceae* from 14 of 18 raw pet food products and zero of 35 non-raw pet food products [57].

There are some limitations to this study. One significant limitation of this study was the collection of data from a single veterinary clinic in Saudi Arabia. The study did, however, provide useful information about the trends in the burden of infections caused by ESBL producers in Saudi veterinary medicine.

## 4. Materials and Methods

### 4.1. Study Design and Sampling

Between January and December 2018, 2000 swabs were collected from 400 cats admitted to a veterinary clinic in Eastern Province of Saudi Arabia. Cats were voluntarily included in the study and were split into two groups according to the reason for visiting the clinic: healthy (*n* = 209) and diseased (*n* = 191). Healthy cats were admitted for vaccination and/or grooming. On the other hand, diseased cats were admitted for clinical examination and displayed one or more of the following clinical symptoms: diarrhea, respiratory signs, conjunctivitis, skin wound, otitis, and/or abscess. A professional veterinarian collected swabs from five different anatomical regions (anus, skin, ear canal, conjunctival sac, and nares) in each cat. Each swab was placed in a sterile tube with 2 mL of liquid brain–heart infusion broth (BHI: Difco) and sent to the laboratory at 4 °C to be analyzed later. Before sample collection, cat owners filled a questionnaire about demographics, cat management, health status, and antimicrobial use. A written consent form was also signed by cat owners who agreed to participate in the study.

### 4.2. E. coli Isolation and Identification

Swabs immersed in 2 mL BHI were preincubated in aerobic conditions overnight at 37 °C. A 10 µL of the suspension was then inoculated onto MacConkey agar (Oxoid, Basingstoke, Hampshire, UK) and incubated under aerobic conditions at 37 °C for 24–48 h. Suggestive *E. coli* colonies were collected, purified on 5% sheep blood agar, Gram stained, and identified to species level using Vitek 2 Compact (BioMerieux, Marcy l’Etoile, France) then stored at −70 °C in 40% glycerol saline. 

For molecular conformation, the QIAamp DNA mini kit (Qiagen SA, Courtaboeuf, France) was used to extract the bacterial DNA from biochemically identified *E. coli* isolates according to the manufacturer’s instructions. PCR was performed on the extracted DNA using primers designed specifically for *E. coli* 16S rRNA amplification and sequencing Weisburg, et al. [58].

### 4.3. Antimicrobial Susceptibility Test

Antimicrobial susceptibility testing was conducted using the disc diffusion method, and the results were interpreted according to Clinical and Laboratory Standards Institute (CLSI) guidelines [59]. In the present study, antimicrobials tested were ampicillin (AMP: 10 μg), cefotaxime (CTX: 30 µg), amoxicillin-clavulanic acid (AMC: 20/10 μg), ciprofloxacin (CIP: 5 μg), tetracycline (TET: 30 µg), gentamicin (GEN: 10 μg), trimethoprim/sulfamethoxazole (SXT: 1.25/23.75 μg), streptomycin (STR: 10 μg), chloramphenicol (CHL: 30 μg), erythromycin (ERY: 15 μg), and meropenem (MEM: 10 μg). Isolates showing resistance to three or more antimicrobials classes (including *β*-lactams as one class) were classified as multidrug-resistant (MDR) [33]. The multiple antibiotic resistance (MAR) index was calculated by dividing the number of antibiotics to which an isolate is resistant to the total number of antibiotics tested for susceptibility [60]. 

### 4.4. ESBL-Producing E. coli Identification

Confirmed *E. coli* isolates were screened for phenotypic ESBL production by disk diffusion tests using cefotaxime (CTX: 30 μg) according to CLSI guidelines [61]. Phenotypically confirmed ESBL-producing *E. coli* were further characterized to identify the virulence (*eaeA*, *hlyA*, *stx1*, and *stx2*) and β-lactamase (*bla*_CTX-M_, *bla*_TEM_ and *bla*_SHV_) genes using primers and reaction conditions (Appendix A) previously described [62,63,64,65].

### 4.5. Statistical Analysis 

The questioner’s data and laboratory analysis data were coded into dichotomous or categorical variables and then combined into a single data set. Data visualization was carried out using R software (version 4.2.0; R Foundation for Statistical Computing, Vienna, Austria). On the other hand, statistical modelling was performed using Stata Statistical Software v.17 (Stata Corp, College Station, TX, USA). Initially, Chi-square and Fisher’s Exact tests were used to identify the association between the presence of *E. coli* isolates and cat condition (healthy vs. diseased), breeds, and sex. However, univariable logistic regression was used to assess the associations between single variables and each of the outcome variables (1- MDR (1 = yes vs. 0 = no) and 2- ESBL (1 = yes vs. 0 = no). Spearman’s rank-order correlation statistics were used to examined for multicollinearity in variables with *p* < 0.25 and before being added as explanatory variables to multivariable logistic regression models. The final model was built manually using backward stepwise elimination at *p* < 0.05. The Hosmer–Lemeshow test was used to assess model fitness; however, receiver operating characteristic curve was used to assess the predictive ability of the model [66].

## 5. Conclusions

*E. coli* isolates recovered from healthy and diseased cats admitted to a veterinary clinic in Saudi Arabia showed high levels of resistance to the majority of tested antimicrobials. Our finding revealed the presence of MDR and ESBL-producing *E. coli* in both healthy and diseased cats, which pose a risk to public and animal health. Thus, to investigate the role of cats as vectors for antimicrobial resistance transmission to humans, an effective antimicrobial stewardship program as well as additional studies using a One Health approach may be required.

## Figures and Tables

**Figure 1 antibiotics-12-00229-f001:**
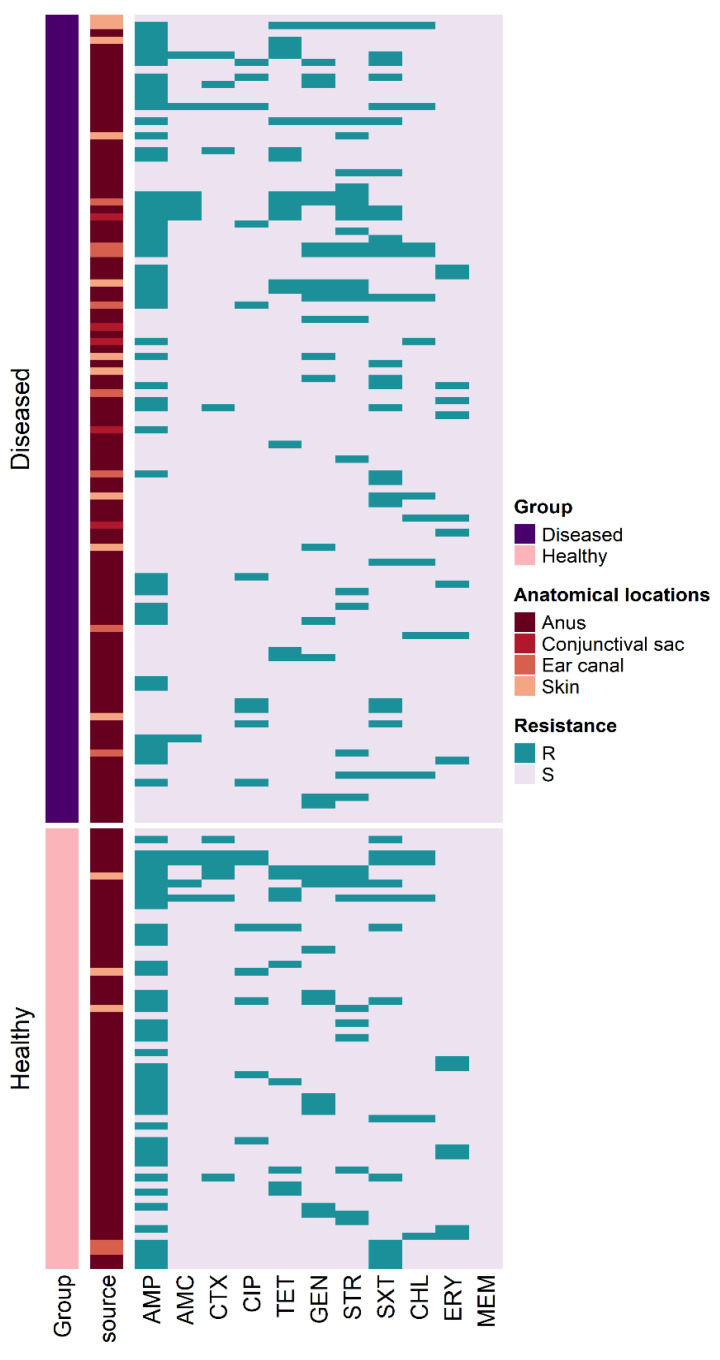
Heat map based on the source and antimicrobial resistance patterns of the 170 *E. coli* isolates found in 400 cats.

**Figure 2 antibiotics-12-00229-f002:**
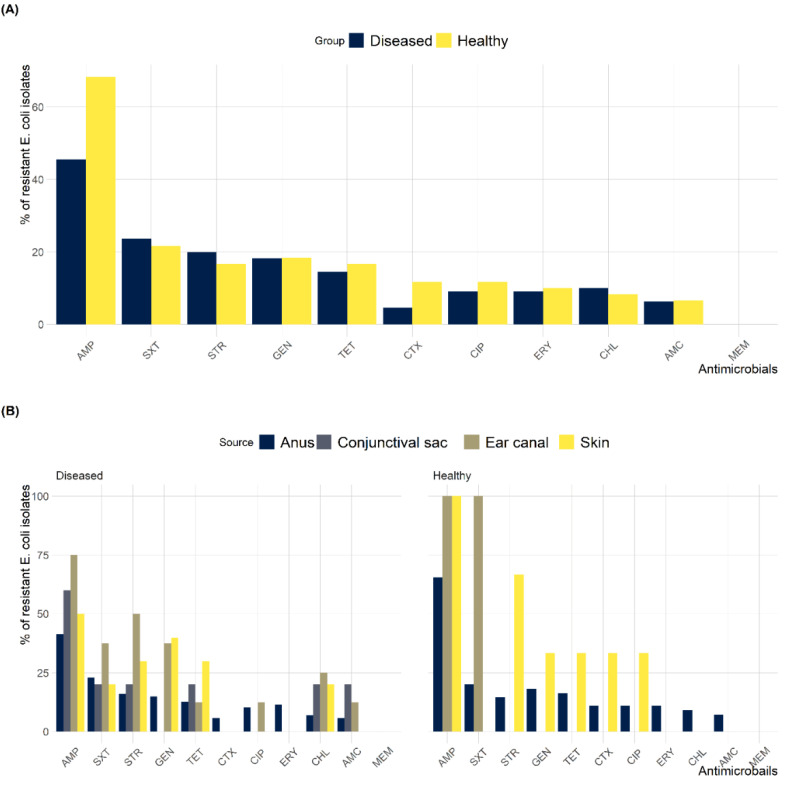
Frequency of antimicrobial resistance of *E. coli* isolates. (**A**) recovered from healthy and diseased cats. (**B**) recovered from different anatomical locations in healthy and diseased cats.

**Figure 3 antibiotics-12-00229-f003:**
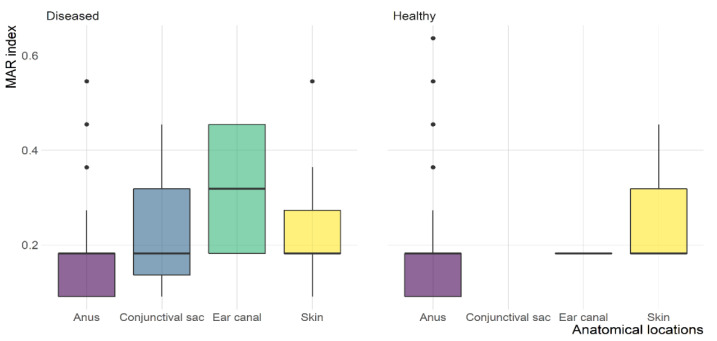
Multiple antibiotic resistance (MAR) index box plot of *E. coli* isolates recovered from different anatomical locations in healthy and diseased cats.

**Table 1 antibiotics-12-00229-t001:** Number of *E. coli* isolates found in 400 cats admitted to a veterinary clinic in Eastern Province of Saudi Arabia.

Anatomical Location	Number (%) of *E. coli* Isolates		Total (*n* = 400)
	Healthy Cats (*n* = 209)	Diseased Cats (*n* = 191)	
Anus	55 (26.3)	87 (45.5)	142 (35.5)
Skin	3 (1.4)	10 (5.2)	13 (3.3)
Ear canal	2 (1.0)	8 (4.2)	10 (2.5)
Conjunctival sac	0 (0.0)	5 (2.6)	5 (1.3)
Nares	0 (0.0)	0 (0.0)	0 (0.0)
Total	60 (28.7)	110 (57.6)	170 (42.5)

**Table 2 antibiotics-12-00229-t002:** Number of multidrug resistance (MDR) and extended-spectrum *β*-lactamase (ESBL) producing *E. coli* isolates found in healthy and diseased cats.

Anatomical Location	N	No. (%) of MDR *E. coli*		Total	No. (%) ESBL *E. coli*		Total
		Healthy	Diseased		Healthy	Diseased	
Anus	142	9 (6.3)	14 (9.9)	23 (16.2)	6 (4.2)	5 (3.5)	11 (7.7)
Skin	13	1 (7.7)	2 (15.4)	3 (23.1)	1 (7.7)	0 (0.0)	1 (7.7)
Ear canal	10	0 (0.0)	3 (30.0)	3 (30.0)	0 (0.0)	0 (0.0)	0 (0.0)
Conjunctival sac	5	0 (0.0)	1 (20.0)	1 (20.0)	0 (0.0)	0 (0.0)	0 (0.0)
Nares	0	0 (0.0)	0 (0.0)	0 (0.0)	0 (0.0)	0 (0.0)	0 (0.0)
Total	170	10 (5.9)	20 (11.8)	30 (17.6)	7 (4.1)	5 (2.9)	12 (7.1)

**Table 3 antibiotics-12-00229-t003:** Resistance, virulence genes and antimicrobial resistance patterns of 12 ESBL-producing *E. coli* isolates found in healthy and diseased cats.

Cat No.	Group	Anatomical Location	Resistance Genes ^1^			Virulence Genes ^1^				Antimicrobial Resistance Patterns	MAR ^2^
			*bla* _CTX-M_	*bla* _TEM_	*bla* _SHV_	*eaeA*	*stx1*	*stx2*	*hlyA*		
3	Healthy	Anus	CTX-M-1	–	–	+	–	–	–	AMP, CTX, SXT	0.27
8	Healthy	Anus	CTX-M-1	+	–	–	–	–	–	AMP, AMC, CTX, CIP, SXT, CHL	0.55
19	Healthy	Anus	CTX-M-15	+	–	–	–	–	–	AMP, AMC, CTX, CIP, SXT, CHL	0.55
25	Diseased	Anus	CTX-M-15	+	–	+	–	+	+	AMP, AMC, CTX, TCY, SXT	0.45
29	Healthy	Anus	CTX-M-15	+	–	–	–	–	–	AMP, CTX, TCY, GEN, STR	0.45
29	Healthy	Skin	CTX-M-15	+	–	–	–	–	–	AMP, CTX, TCY, GEN, STR	0.45
41	Diseased	Anus	CTX-M-15	–	–	+	–	–	–	AMP, CTX, GEN,	0.27
44	Healthy	Anus	CTX-M-15	+	–	–	–	–	–	AMP, AMC, CTX, TCY, STR, SXT, CHL	0.64
53	Diseased	Anus	CTX-M-1	–	+	–	–	–	–	AMP, AMC, CTX, CIP, SXT, CHL	0.55
82	Diseased	Anus	CTX-M-1	–	–	–	–	–	–	AMP, CTX, TCY	0.27
216	Diseased	Anus	CTX-M-15	–	–	+	–	+	+	AMP, CTX, SXT	0.27
267	Healthy	Anus	CTX-M-15	–	–	+	–	–	–	AMP, CTX, SXT	0.27

^1^ + = resistance or virulence genes positive; – = resistance or virulence negative; ^2^ MAR = multiple antibiotic resistance index.

**Table 4 antibiotics-12-00229-t004:** Univariable results for risk factors associated with multidrug resistance (MDR) and extended-spectrum *β*-lactamase (ESBL) producing *E. coli* isolates found in 400 cats.

Factors	MDR *E. coli*		ESBL *E. coli*	
	OR ^1^	*p*-Value	OR ^1^	*p*-Value
Family use antimicrobials				
No	1.00 (ref.)		1.00 (ref.)	
Yes	13.1	0.000	9.7	0.001
Family member with acne				
No	1.00 (ref.)		1.00 (ref.)	
Yes	3.3	0.004	0.89	0.870
Hospitalization				
No	1.00 (ref.)		1.00 (ref.)	
Yes	3.1	0.009	1.9	0.337
Previous antimicrobials use for cat				
No	1.00 (ref.)		1.00 (ref.)	
Yes	3.4	0.006	4.1	0.040
Current antimicrobials use for cat				
No	1.00 (ref.)		1.00 (ref.)	
Yes	8.3	0.000	-	-
Child at home				
No	1.00 (ref.)		1.00 (ref.)	
Yes	2.4	0.052	2.9	0.122
Cat Living				
Indoor	1.00 (ref.)		1.00 (ref.)	
Indoors–outdoors	0.7	0.461	2.3	0.295
Reason being at clinic				
Vaccination and/or grooming	1.00 (ref.)		1.00 (ref.)	
Treatment	2.03	0.099	0.91	0.877
Cat care				
Adult male	1.00 (ref.)	0.000	1.00 (ref.)	0.016
Adult female	0.15	0.000	0.21	0.025
Child	-	-	-	-
All family	0.15	0.013	-	-
Food type				
Dry	1.00 (ref.)	0.003	1.00 (ref.)	0.006
Wet	0.99	0.992	2.5	0.456
Raw	4.2	0.013	16.8	0.001
Home available	5.2	0.001	12.1	0.005

^1^ OR: odds ratio.

**Table 5 antibiotics-12-00229-t005:** Multivariable results for risk factors associated with multidrug resistance (MDR) and extended-spectrum *β*-lactamase (ESBL) producing *E. coli* isolates found in 400 cats.

Factors	MDR *E. coli*		ESBL *E. coli*	
	OR (95% CI) ^1^	*p*-Value	OR (95% CI) ^1^	*p*-Value
Family use antimicrobials				
No	1.00 (ref.)		1.00 (ref.)	
Yes	20.0 (6.29–63.69)	0.000	16.6 (3.29–84.08)	0.001
Previous antimicrobials use for cat				
No	1.00 (ref.)		1.00 (ref.)	
Yes	5.5 (1.83–16.36)	0.002	7.8 (1.66–36.34)	0.009
Child at home				
No	1.00 (ref.)			
Yes	3.4 (1.14–10.30)	0.027	-	-
Cat care				
Adult male	1.00 (ref.)	0.019	-	-
Adult female	0.20 (0.06–0.69)	0.011	-	-
Child	-	-	-	-
All family	0.21 (0.03–1.29)	0.092	-	-
Food type				
Dry	1.00 (ref.)	0.006	1.00 (ref.)	0.001
Wet	0.93 (0.15–5.64)	0.939	2.7 (0.22–33.50)	0.431
Raw	8.5 (1.74–41.70)	0.008	59.7 (7.16–497.54)	0.000
Home available	6.3 (1.76–22.80)	0.005	12.4 (1.94–79.63)	0.008
_cons	0.004 (0.001–0.020)	0.000	0.001 (0.0001–0.005)	0.000

^1^ OR: odds ratio; CI: confidence interval.

## Data Availability

The data presented in this study are available on request from the corresponding author.

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
