# Peer review of "Prevalence and Risk Factors Associated with Multidrug Resistance and Extended-Spectrum β-lactamase Producing E. coli Isolated from Healthy and Diseased Cats"

_antibiotics, 2023, doi:10.3390/antibiotics12020229_

Round 1
Reviewer 1 Report
da tradução
Material and Methods In the methodology, did the animals that were collected samples other than the anal swab have infections in these places? Because, it is obvious a larger amount of E. coli isolated in the rectal swab.It would be interesting to sequence the TEM and SHV genes, as TEM-1 and TEM-2, as well as SHV-1 are cephalosporinases, but not extended spectrum, not constituting an ESBL. The virulence factors are random, with no textual connection with the rest of the work, it would be interesting to discuss a little in the introduction. In materials and methods, you talk about the use of Baird Parker culture medium, which is a selective medium for Staphylococcus, what was the purpose of its use? It got pretty confusing. Why was CLSI 2022 not used? The antimicrobial erythromycin has action against Gram positive bacteria, what was the purpose of using it in the study?
Author Response
R1.1. Material and Methods In the methodology, did the animals that were collected samples other than the anal swab have infections in these places? Because, it is obvious a larger amount of E. coli isolated in the rectal swab.
AU: Yes, you're right. The majority of the E. coli isolates (142/170) in this study were recovered from rectal swabs. This result was not surprising and was consistent with previously published work. However, as shown in Table 1, 28 E. coli isolates were recovered from other anatomical regions.
R1.2. It would be interesting to sequence the TEM and SHV genes, as TEM-1 and TEM-2, as well as SHV-1 are cephalosporinases, but not extended spectrum, not constituting an ESBL.
AU: The reviewer makes an excellent point, and we agree with him. The original ESBL enzymes were TEM and SHV variants, and to date, 243 TEM variants and 228 SHV variants have been identified, although not all of these have the ESBL phenotype. Since the first reports of CTX-M in the late 1980s, the worldwide spread of isolates carrying these ESBLs has been referred as the 'CTX-M pandemic'. CTX-M-type enzymes have been recognized as the most common ESBL group since the early 2000s, replacing TEM and SHV as the dominant ESBL type. Although TEM- and SHV-type ESBLs are still encountered, their presence in clinical isolates appears to have little impact.
In a recent survey of European isolates, TEM-type ESBLs were detected in less than 1% of ESBL-producing E. coli and Klebsiella pneumoniae (Kazmierczak et al., 2020). All isolates identified as ESBLs phenotypically in this survey were positive for the CTX-M gene, with varying results for the TEM and SHV genes. So, based on the recent literature and our findings, we hypothesized that the ESBL activity in our isolates was due to the presence of CTX M enzymes. Nonetheless, the authors agree with the reviewer that sequencing of the TEM and SHV genes would be interesting and warranted in future research.
- Kazmierczak, K.M., de Jonge, B.L., Stone, G.G. and Sahm, D.F., 2020. Longitudinal analysis of ESBL and carbapenemase carriage among Enterobacterales and Pseudomonas aeruginosa isolates collected in Europe as part of the International Network for Optimal Resistance Monitoring (INFORM) global surveillance programme, 2013–17. Journal of Antimicrobial Chemotherapy, 75(5), pp.1165-1173.
R1.3. The virulence factors are random, with no textual connection with the rest of the work, it would be interesting to discuss a little in the introduction.
AU: Done (lines 59-65).
R1.4. In materials and methods, you talk about the use of Baird Parker culture medium, which is a selective medium for Staphylococcus, what was the purpose of its use? It got pretty confusing.
AU: Thank you for noting that. This media was included by mistake in the lab book under E. coli and subsequently in the manuscript as these samples were also cultured to isolate staphylococcus spp.. The sentence was corrected to make clear for the readers (Lines 281-282).
R1.5. Why was CLSI 2022 not used?
AU: When we interpreted our results, the CLSI 2022 had not yet been published. However, when we looked up the CLSI M100-ED32:2022 Performance Standards for Antimicrobial Susceptibility Testing, 32nd Edition, we did not find any modification to the zone diameter for Enterobacterales regarding the antibiotics used in this work.
R1.6. The antimicrobial erythromycin has action against Gram positive bacteria, what was the purpose of using it in the study?
AU: Again, reviewer raises a good point. We aimed to investigate the susceptibility of E. coli isolates to the macrolides. The azithromycin discs were unavailable, so we used erythromycin as a representative of the macrolides. Furthermore, In Saudi Arabia, erythromycin commonly used in companion animals and humans and several studies isolated E. coli with high resistance rate to erythromycin.
- Alharbi, N.S., 2020. Escherichia coli in Saudi Arabia: An Overview of Antibiotic-Resistant Strains. Biosciences Biotechnology Research Asia, 17(3), pp.443-457.
- Abo-Amer, A.E., Shobrak, M.Y. and Altalhi, A.D., 2018. Isolation and antimicrobial resistance of Escherichia coli isolated from farm chickens in Taif, Saudi Arabia. Journal of global antimicrobial resistance, 15, pp.65-68.
Reviewer 2 Report
This study investigated E. coli from both healthy and diseased cats in Eastern Province of Saudi Arabia. The results showed the presence of MDR and ESBL-producing E. coli in both healthy and diseased cats.
1. The methods used to identify the virulence (eae, hlyA, stx1, and stx2) and β-lactamase (blaCTX-M, blaTEM and blaSHV) genes were adopted from very previous papers. Many stx1/stx2 subtypes or other gene variants were identified since then. It is not sure these primers covering these new subtypes or not.
2. Among all cats (n = 400), 165 were found to carry E. coli and a total of 170 E. coli isolates were found. How many isolates from each sample were kept for further analyzing? If more than one isolate from a single sample was used, the frequency should be biased.
Author Response
R2.1. This study investigated E. coli from both healthy and diseased cats in Eastern Province of Saudi Arabia. The results showed the presence of MDR and ESBL-producing E. coli in both healthy and diseased cats. The methods used to identify the virulence (eae, hlyA, stx1, and stx2) and β-lactamase (blaCTX-M, blaTEM and blaSHV) genes were adopted from very previous papers. Many stx1/stx2 subtypes or other gene variants were identified since then. It is not sure these primers covering these new subtypes or not.
AU: The primers used in this study were chosen based on the study objectives. In terms of virulence genes, our goal was to identify the pathotypes of the isolates that exhibited extended spectral beta-lactamase activity in order to highlight their public health significance. Shiga toxins (Stx) are classified into two antigenic types: Stx1 and Stx2. Stx1 and Stx2 variants are classified into three (Stx1a, Stx1c, and Stx1d) and seven (Stx2a, Stx2b, Stx2c, Stx2d, Stx2e, Stx2f, and Stx2g) subtypes, respectively. This subtyping scheme is useful in assessing the potency of these toxins. However, in this study we did not aim to study the potency of thesis toxins against cell lines. Our aim was to figure out if there are any enteropathogenic or enterohaemorrhagic pathotypes among the ESBL isolates.
R2.2. Among all cats (n = 400), 165 were found to carry E. coli and a total of 170 E. coli isolates were found. How many isolates from each sample were kept for further analyzing? If more than one isolate from a single sample was used, the frequency should be biased.
AU: The reviewer makes an excellent point, but we'd like to clarify that we reported two different frequencies (cat level and isolate level frequencies). The cat level frequency, E. coli was isolated from 165 of 400 cats, resulting in a 41.3% isolation frequency, with 28.2% (59/209) healthy and 55.5% (106/191) diseased cats. The total number of E. coli isolates was 170 because E. coli was isolated from two different anatomical locations in five cats. the frequency of resistance and susceptibility were calculated at the isolate level.
Round 2
Reviewer 2 Report
1. Former Q1, It is not sure these primers covering these new subtypes or not. please check.
2. Former Q2, It should be explained in the revised manuscript.
Do not just argue.
Author Response
R2.1. Former Q1, It is not sure these primers covering these new subtypes or not. please check.
AU: The selected primer can cover the variants of the amplified genes. The primer used to amplify the eae gene can cover the subtypes (α, β, δ, and γ). The primer used to amplify the stx genes were designed to flank the region of the A subunit of stx genes containing the codon corresponding to glutamic acid 166 in stx-II (amino acid 167 in stx-I). This region is relatively conserved among the SLT genes and can amplify most of the stx toxin subtypes.
R2.2. Former Q2, It should be explained in the revised manuscript.
AU: Explained in the manuscript as recommended (Lines 104-105).